# THREE-HEAD NEURAL NETWORK ARCHITECTURE FOR ALPHAZERO LEARNING

## ABSTRACT

The search-based reinforcement learning algorithm AlphaZero has been used as a general method for mastering two-player games Go, chess and Shogi. One crucial ingredient in AlphaZero (and its predecessor AlphaGo Zero) is the two-head network architecture that outputs two estimates — policy and value — for one input game state. The merit of such an architecture is that letting policy and value learning share the same representation substantially improved generalization of the neural net. A three-head network architecture has been recently proposed that can learn a third action-value head on a fixed dataset the same as for two-head net. Also, using the action-value head in Monte Carlo tree search (MCTS) improved the search efficiency. However, effectiveness of the three-head network has not been investigated in an AlphaZero style learning paradigm. In this paper, using the game of Hex as a test domain, we conduct an empirical study of the three-head network architecture in AlpahZero learning. We show that the architecture is also advantageous at the zero-style iterative learning, producing neural network models stronger than those from the two-head counterpart in the same MCTS.

## 1 INTRODUCTION

Computational advancements of deep neural networks (Bengio, 2009; LeCun et al., 2015) have led to progresses in a number of scientific areas, ranging from image classification (Krizhevsky et al., 2012), speech recognition (Graves et al., 2013), natural language processing (Collobert et al., 2011), playing Atari games (Mnih et al., 2015), to the prominent achievement of AlphaGo (Silver et al., 2016), AlphaGo Zero (Silver et al., 2017b), and AlphaZero (Silver et al., 2017a; 2018) learning systems for playing multiple classic two-player alternate-turn perfect-information zero-sum games — games of which kind have been a research object as old as the subject of artificial intelligence itself. Arguably, the accomplishments of AlphaGo and its successors represent a result of continual effort from two lines of research directions. The *heuristic search* (or sometimes *AI planning*) community studies various intelligent search strategies (Pearl, 1984) (e.g., *best-first* search) given that a space of states can be generated gradually by description of the problem (e.g., rules of Go), the goal is to find a desired solution using smaller computational cost by exploring promising regions preferably; the study of machine learning attempts to summarize a general concept from experience data, therefore enabling probably correct prediction even in unseen scenarios.

The combination of both learning and search techniques dates back to Samuel's studies in checkers (Samuel, 1959), where two learning methods, *rote learning* and *generalization learning*, were studied and integrated into a look-ahead minimax search; the later development leveraging advancements from both learning and search techniques (e.g., training on games produced by stronger players with more expressive function approximation for better generalization learning and using better $\alpha\beta$ pruning (McCarthy et al., 2006) in search) achieved better results (Samuel, 1967). Noticeably, the learning techniques are only used to construct an evaluation function for checkers positions from historical playing records either generated by human or computer players. However, in certain games (e.g., chess) where a large body of human players exist, manually constructing a heuristically correct evaluation function is also possible. Therefore, rather than attempting to automatic construct an evaluation functions, much early research effort focus on analyzing formal properties of search algorithms given certain assumption on the correctness of an existing evaluation function (Knuth & Moore, 1975; Pearl, 1980a), proposing new search routines that may lead to more efficient pruning (Pearl, 1980b; Plaat et al., 1996), or devising procedures when the assumed correctness of an

evaluation failed to consign in practice (Harris, 1975). Indeed, optimizing using minimax upon an approximated evaluation has been regarded as a "deadly sin" in the eye of statistics, since such an optimization could only compound the approximation error, therefore resulting pathological behavior of search (Nau, 1982), i.e., the deeper the search the worse of the obtained evaluation. However, in practice, pathological behavior was seldom observed in a number of games. Pearl (1983) presented several observations that may explain the phenomenon, such as the approximation error tends to decrease as search goes deeper, the ubiquitous of perfect evaluation in some branches of the search, the fact that the search space is a directed acyclic graph rather than a tree, and the relatively small branching factor of the game of interest. Game-searching by minimax continued to triumph as computers become faster and more general or game-specific searching techniques are introduced, culminating to the successes of champion playing strength against human players in games such as checkers (Schaeffer et al., 1992), chess (Campbell et al., 2002), Othello (Buro, 1998) and backgammon (Tesauro, 1995).

It follows that game AI research began to switch attention on several more challenging games where reliable human-directed evaluation function is difficult to construct and the branching factor is significantly larger, such as the game of Go (Müller, 2002) and the game of Hex (Van Rijswijck, 2002a). Coulom (2006) proposed a Monte Carlo tree search (MCTS) for computer Go; it uses Monte Carlo simulation to sidestep the need of a static evaluation function. From the heuristic search perspective, MCTS can be regarded as an instance of the general best-first search depicted by Pearl (1984). Different from $\alpha\beta$ pruning, which relies on an accurate move ordering for preferable exploration, best-first search iteratively adjusts its search effort by summarizing the new information backed from leaf evaluations, at the expense of explicitly storing the whole search tree in memory. To select a leaf node to expand and evaluation from the search tree, a UCT search can be used (Kocsis & Szepesvári, 2006) for a balance between exploration and exploitation. Notably, unlike $\alpha\beta$ minimax search, MCTS does not apply hard-min or max when backing up approximated leaf evaluation but uses average to summarize all previous evaluations while preserving asymptotic convergence. The superiority of MCTS is also manifested by its flexibility of integrating learned heuristic knowledge into the search (Gelly & Silver, 2007). That is, by incorporating an existing move-ranking function for better leaf node selection (i.e., progressive bias by Chaslot et al. (2008) or PUCT by Rosin (2011)), the search can focus on only promising moves therefore effectively reducing the high branching factor; by leveraging more informed pattern-based Monte Carlo playout (Coulom, 2007), better leaf evaluation can be obtained and backed up therefore further enforcing the best-first behavior, although in practice stronger playout policy for leaf evaluation does not always result in stronger performance given the same finite computation of MCTS (Silver & Tesauro, 2009).

Continual effort were made towards adding more accurate prior knowledge to MCTS. The successes of deep neural networks in other AI domains, such as image classification (Krizhevsky et al., 2012), revived the application of neural networks for approximate human's evaluation in Computer Go (Maddison et al., 2015; Clark & Storkey, 2015; Tian & Zhu, 2015), leading to the development of AlphaGo Fan (Silver et al., 2016) which uses policy network for providing prior knowledge in move selection and value network for leaf node evaluation in MCTS — it became the first program defeating human professional player in the game of Go. Subsequently improved versions AlphaGo Lee defeated a more pronounced world champion player in public match. A even stronger version called AlphaGo Master was strong enough that it never lost any game against any top human players in a set of online match. The commonality of these programs is that the neural networks used in search were all first trained on human professional game records. Later, it was found that even without using human data to initialize the neural network weights, a closed loop learning paradigm could still learn MCTS players stronger than AlphaGo Lee and AlphaGo Master (after doubling neural net size); the paradigm is named as AlphaGo Zero (Silver et al., 2017b). By further removing other Go specific encoding in AlphaGo Zero, a more general version — AlphaZero (Silver et al., 2017a; 2018) — was applied to chess, Shogi and Go, producing strong players in all these games after separately training on each of them. We summarize the major differences between AlphaGo Master, AlphaGo Zero and AlphaZero in Figure 1.

However, one limitation of the Zero algorithms (AlphaGo Zero and AlphaZero) is that a large computation cost could be needed because they use search to produce millions of training games, where each move in a game was produced by calling neural networks hundreds of times for building the search-tree — AlphaZero used thousands of TPUs in its week-long training on the game of Go (Silver et al., 2018). A three-head neural network architecture was proposed in (Gao et al., 2018b)

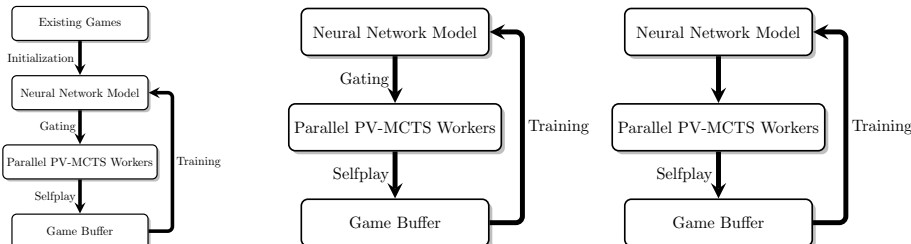

(a) AlphaGo Master (Silver et al., 2017b; 2016)    (b) AlphaGo Zero (Silver et al., 2017b)    (c) AlphaZero (Silver et al., 2017a; 2018)

Figure 1: Flowcharts for AlphaGo Master, AlphaGo Zero and AlphaZero; each of them is a simplification of its predecessor; that is, AlphaGo Zero removes the need of human data in training, while in AlphaZero *gating* is further discarded (gating refers to playing tournament using existing neural models in PV-MCTS thus a model parameter will passed down to selfplay workers only when its win-percentage exceeds a certain threshold, e.g., 55%). Notice that AlphaGo Zero and AlphaZero used asynchronous training, i.e., selfplay game generation and neural network training were conducted simultaneously given a sufficient number of games have been stored in the buffer. Detailed differences on search and learning, such as hyperparameters, are omitted in these diagrams.

which can learn an additional action-value head on search-produced training dataset, akin to the two-head network used in AlphaGo Master, AlphaGo Zero and AlphaZero. In this paper, we conduct an empirical comparative study on AlphaZero style learning using two- and three-head neural networks. We find that, in comparison to the two-head counterpart, three-head network can also lead to improvement in AlphaZero style learning paradigm.

## 2 ANATOMY OF ALPHAGO ZERO AND ALPHAZERO

We provide an anatomy to AlphaGo Zero and AlphaZero, along with discussion with their predecessors AlphaGo Fan, AlphaGo Lee and AlphaGo Master. See Appendix A for preliminaries on Monte Carlo tree search.

### 2.1 POLICY VALUE MONTE CARLO TREE SEARCH

MCTS is an essential component in AlphaGo Zero and AlphaZero since it is the algorithm upon which training data for the neural network are produced; that is, the policy network is trained by mimicking the move probabilities generated by MCTS, while the value network is to learn to predict game result generated by MCTS players.

A two-head policy-value network $f_\theta$ provides two estimates given one input game-state, i.e., $\boldsymbol{p}, v \leftarrow f_\theta(s)$ where $s$ is a game-state, $\boldsymbol{p}$ is a policy over actions at $s$ and $v$ is an estimate of the optimal state-value of $s$. After one evaluation, policy and value estimates are added to the Monte Carlo tree search (PV-MCTS) in the following way. At each iteration of MCTS, after a leaf node $s$ is selected, the game state of that leaf node is then passed to neural net for inference, $\boldsymbol{p}, v \leftarrow f_\theta(s)$. The value estimate $v$ is backpropaged to the search tree. The leaf node is subsequently expanded, for each action $a$ from $s$, the newly created child node stores: $\{N(s,a) = 0, W(s,a) = 0, Q(s,a) = 0, P(s,a) = \boldsymbol{p}_a\}$, where $N(s,a)$ the visit count of $a$ at $s$, $W(s,a)$ is the number of wins, $Q(s,a)$ is the accumulated action-value (equivalent to negated $V(s')$ is acting $a$ at $s$ results $s'$), and $P(s,a)$ is the prior probability for action $a$. AlphaGo Fan used an expansion threshold $n_{th} > 0$, therefore a leaf node is expanded only when it has been visited more than $n_{th}$ times. AlphaGo Zero and AlphaZero used an expansion threshold of $0$ because for each selected leaf node, to back up an estimate, neural network has to be called; however, the two-head neural network provides two estimates $\boldsymbol{p}$ and $v$; naturally, to not waste GPU computation, probabilities in $\boldsymbol{p}$ has to be saved in newly created child nodes by node expansion.

| Search Param. | Description | Learning Param. | Description |
|---|---|---|---|
| $n_{mcts}$ | num of MCTS iterations | $d_{nn}$ | depth of neural net |
| $c_{puct}$ | PUCT coefficient | $w_{nn}$ | width of neural net |
| $\alpha$ | root dirichlet noise parameter | $\beta$ | initial learning rate |
| $\eta$ | move select dithering threshold | - | how to schedule learning rate |
| $\tau$ | dither temperature | $c$ | $L_2$ regularization in loss function |
| - | reuse subtree or not? | $B$ | mini-batch size |
| $\epsilon$ | root fraction for dirichlet noise | $I_{nn}$ | input planes to neural net |

Table 1: Search and Learning parameters in AlphaGo Zero and AlphaZero. In AlphaGo Zero, $n_{mcts} = 1600$ while in AlphaZero $n_{mcts} = 800$. PUCT coefficient in AlphaGo Zero were not explicitly specified, studies by Tian et al. (2019) suggests $c_{puct} = 1.5$. $\alpha = 0.03$ in Go; dither selection threshold $\eta$ of MCTS was 30 with temperature 1.0; weight for dirichlet noise $\epsilon = 0.25$. Note that in (Silver et al., 2018), a dynamic PUCT coefficient with base 1.25 was used; however, as noted in (Silver et al., 2018), during training, due to small visit count of each game state, the coefficient is almost static.

## 2.2 CLOSED LOOP LEARNING

As in Figure 1, AlphaGo Zero and AlphaZero can be summarized as a closed loop learning paradigm. The predecessor AlphaGo Master might also be called closed loop learning based on description revealed in (Silver et al., 2017b). It begins by initializing neural network weights either with supervised learning on human data or just with random weights. Then, the neural network is plugged into an MCTS and passed to a number of parallel workers for producing selfplay games. These games are stored in an external buffer, and subsequently an updated neural network model is produced by retraining the neural network on game examples sampled from the buffer. Even though AlphaGo Zero and AlphaZero were run in asynchronous manner, our summarization in Figure 1 naturally suggests a synchronous paradigm, where at each iteration the learning worker should wait until all selfplay worker to finish all their games before optimizing the neural network. Whether the asynchronous implementation is superior to the synchronous execution, or how advantageous it could be, has never been formally investigated. On the other side, asynchronicity introduces an additional difficulty for reproducibility (Henderson et al., 2018) due to the fact that executing results are dependent on the relative speed of the selfplay and learning workers in that specific execution. We therefore believe that the synchronous implementation of AlphaZero is worth investigating as it has never been formally investigated before. For example, in model-free reinforcement learning algorithm A3C (Mnih et al., 2016), asynchronicity was conjectured as one reason for the observed good performance, other implementation found that the synchronous version A2C (OpenAI, 2017) can achieve equivalent results as A3C.

## 2.3 SEARCH AND LEARNING HYPERPARAMETERS

Due to the possibility of different hyperparameter choices, AlphaZero represents a family of algorithms, rather than a single and rigidly defined procedure. We classify the hyperparameters into three categories: (1) search parameters, (2) learning parameters, (3) execution parameters. While (3) could be specified by referring to the specific game and available computation hardware, we summarize the (1) and (2) in Table 1. Observing that the learning parameters are in large accordance with many other deep learning applications (Goodfellow et al., 2016), we emphasize that search hyperparameters used to configure MCTS are not typically employed in heuristic search, making the setting of these parameters difficult.

## 2.4 THEORETICAL JUSTIFICATION

In terms of reinforcement learning, the algorithmic framework of AlphaGo Zero and AlphaZero has been summarized as an *approximated policy iteration* (Bertsekas, 2011) (or generalized policy iteration as in (Sutton & Barto, 2017)), where the *policy improvement* is due to running PV-MCTS at each state, and *policy evaluation* is achieved by regression with respect to the search-based selfplay game result (Silver et al., 2017b). However, such an interpretation is essentially high-level: It does not provide detailed explanation on convergence properties of the algorithm, nor does it give explicit

and practical guidance on how to select better hyperparameters for search, neural network design and training. Another interpretation names the whole framework as an Expert Iteration (ExIt) (Anthony et al., 2017) by drawing connection to Systems I and II psychological theories on human mind — the *fast* intuition by deep neural network and the *slow* thinking by tree search (Evans & Stanovich, 2013). The same drawback remains; ExIt does not provide detailed guidance on hyperparameter choices either.

## 3 POTENTIAL MERITS OF THREE-HEAD NEURAL NETWORK ARCHITECTURE

A three-head network $\theta$ provides three estimates from three heads: $\boldsymbol{p}, \boldsymbol{q}, v \leftarrow f_\theta(s)$. The additional $\boldsymbol{q}$ head is simply a two-layer network appended to the common representation shared by $\boldsymbol{p}$ and $v$ heads: the first layer uses a single $1 \times 1$ convolution filter, and the second layer uses a layer of $N \times N$ fully-connected units with $\tanh$ function to transform the output into action values. For brevity, we shall call three-head network as 3HNN, and 2HNN for the two-head architecture. A 3HNN can be optimized using the following loss function:

$$L(f_\theta; \mathcal{D}) = \sum_{(s,a,z_s) \in \mathcal{D}} \left( \left( (z_s - v(s))^2 + \underbrace{(z_s + q(s,a))^2}_{R1} + \underbrace{\frac{\max(-z_s, 0)}{|\mathcal{A}(s)|} \sum_{a' \in \mathcal{A}(s)} (z_s + q(s,a'))^2}_{R2} \right. \right.$$
$$\left. \left. + \underbrace{(\min_{a'} q(s,a') + v(s))^2}_{R3} \right) - \boldsymbol{\pi}^T(s) \log \boldsymbol{p}(s) + c||\theta||^2 \right) \quad (1)$$

where $c$ is a constant for the level of $L_2$ regularization; $\mathcal{D}$ is a set of game examples produced by search; $(s, a)$ is a state-action pair from example set $\mathcal{D}$; $z_s$ is the game result with respect to the player to-play at $s$ that can be extracted from $\mathcal{D}$; and $\mathcal{A}(s)$ is the set of actions at $s$. In comparison to the loss function used in training 2HNN, three additional value error functions, $R1$, $R2$ and $R3$, are added. $R1$ can be extracted from $\mathcal{D}$ directly; $R2$ is produced by a data-augmentation method; and $R3$ is an optimal consistence penalty between state- and action-value heads; see (Gao et al., 2018b). We present the following potential merits that three-head neural network may bring to MCTS and AlphaZero learning.

- More efficient MCTS. One advantage that has been exploited by 3HNN in (Gao et al., 2018b) is allowing node expansion threshold $n_{th} > 0$. Because when a new child node is first created, like the $P(s,a) = \boldsymbol{p}_a$, the action-value can be initialized to $Q(s,a) = \boldsymbol{q}_a$. Therefore, in future MCTS iteration, a leaf value estimate can be obtained simply by querying $Q(s,a)$ without expanding and evaluating the node using $f_\theta$. When a node is expanded, state-value head is backed up immediately. This improves the search efficiency of MCTS as it allows more iterations of search given the same computation budget.

- Auxiliary task for better neural net learning. Yet how deep neural networks manage to generalize is much unknown except some intuitive explanation (Zhang et al., 2017), many empirical evidence seem to suggest that adding auxiliary tasks (Jaderberg et al., 2016) during neural net training generally produces better results. In AlphaGo, it was found that simply combining policy and value network learning into a two-head architecture produces an 600 Elo gain (Silver et al., 2017b).

- Improved data efficiency. Using 3HNN also improves the data efficiency in the sense that on the same dataset, one more estimator is learned, and if such an advantage is fully used in search, it is expected that few number of training samples might be required to achieve similar results as using 2HNN.

## 4 EXPERIMENTS

In this section, we present experiments conducted on the game of Hex. We first give a brief review on previous research in this game, then describe the testing data we used to measure learning progress.

Finally, we present experiment results and comparisons of 2HNN and 3HNN in the same AlphaZero learning.

## 4.1 THE GAME OF HEX

The game of Hex (Hayward & Toft, 2019) can be played on arbitrary $N \times N$ hexagonal board, where two players Black and White each is given two opposing borders; at each move, one player places a stone of their color on the board. Usually, Black starts first, then the game continues in alternate turns; the goal for each player is to form a monochromatic chain connecting their opposing borders. The game became well-known after Nash (1952), who proved that there is a first-play winning strategy for any board size $N \times N$ though explicit winning strategies are unknown.

The game was first studied from an AI perspective in 1950s by Shannon (1953), who designed an electronic resistance simulated evaluation to play Hex — after observing that devising an evaluation using expert knowledge could be difficult in this game (unlike, e.g., chess). However, progresses have been slow in developing strong algorithms for the game. Using a two-distance evaluation extracted from graph properties of Hex, together with advanced implementation of $\alpha\beta$ search, in late 1990s Queenbee (Van Rijswijck, 2002b) became the first program achieving novice strength on small board sizes. The discovery of a hierarchical algebra (implemented as H-Search (Anshelevich, 2002)) to bottom up build connection strategies resulted moderate strong playing players (e.g., Hexy (Anshelevich, 2000) and Six (Hayward, 2006)) using $\alpha\beta$ search. The benefits of H-Search are manifold: (1) it can be used for early endgame state detection; (2) it often induces significant pruning due to *mustplay* (Hayward et al., 2004); and (3) it induces a simplified Hex graph where more accurate evaluations — e.g., electronic resistance — can be extracted from. Yet, as suggested in (Berge, 1981; Hayward & Van Rijswijck, 2006), it is possible to identify more pruning by manually analyzing the graph properties of Hex; the study of inferior cells are towards this end. Notably, both pruning from H-Search and inferior cells are provably correct knowledge to the game, thus optimal strategy is always preserved. Continual effort in identifying more local inferior patterns (Henderson, 2010), better H-Search implementations (Pawlewicz et al., 2015), and better parallel exhaustive search (Pawlewicz & Hayward, 2013) have subsequently led to board sizes from $7 \times 7$ (Hayward et al., 2004), $8 \times 8$ (Henderson et al., 2009), and $9 \times 9$ (Pawlewicz & Hayward, 2013) being *solved* [1]. These knowledge computation have been implemented in an open source codebase `benzene` (Arneson et al., 2010b). For playing, applying H-search and inferior cells improves the application of MCTS to the game, resulting program MoHex (Arneson et al., 2010a) (available inside of `benzene`). Like Go, later improved version MoHex 2.0 (Huang et al., 2013; Pawlewicz et al., 2015) add more accurate heuristic knowledge to in-tree selection and leaf evaluation by pattern-based Monte Carlo playout. Recent MoHex versions further incorporate knowledge due to neural networks in its node selection and evaluation (Gao et al., 2017a; 2018b) — adding neural network for both player and solver leads to `neurobenzene` where how to train these neural nets is configurable, and we shall use this platform to conduct our study.

## 4.2 IMPLEMENTATION

We focus on small board size $9 \times 9$ Hex, taking advantage of the existing solver (Henderson, 2010; Pawlewicz et al., 2015; Pawlewicz & Hayward, 2013) to measure the progress of AlphaZero learning. However, since H-search and inferior cell analysis could induce too much pruning on such small board size, we configure MoHex to turn off these knowledge computation during training, simulating a pure MCTS player whose performance is majorly influenced by the quality of the neural net knowledge learned from playing data. We use a synchronous implementation that follows the rightmost workflow in Figure 1. We use the neural network architecture exactly as in (Gao et al., 2018b), consisting of 10 residual blocks where each block is built with two convolutional layers each is with $32\ 3 \times 3$ filters. See Appendix B for implementation details and parameter choices.

## 4.3 TEST DATA

To measure the progress of neural net learning, we use two datasets.

---

[1]Note that solving $9 \times 9$ requires months of parallel computation

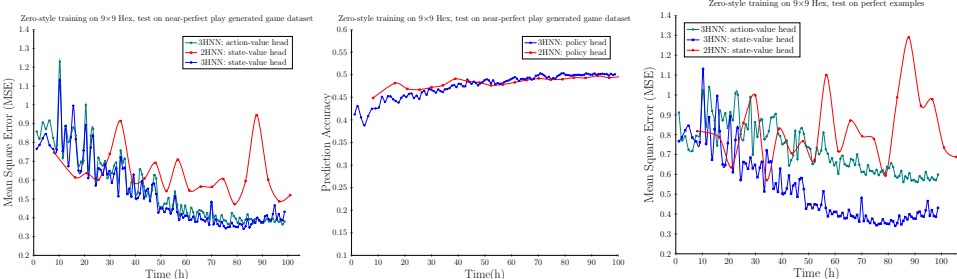

Figure 2: AlphaZero-2HNN versus AlphaZero-3HNN learning progresses measured in training time. MCTS-3HNN performs 800 iteration MCTS faster than MCTS-2HNN because it uses the default expand threshold of 10, while MCTS-2HNN uses 0. Consequently, 3HNN leads to faster AlphaZero learning than that of 2HNN. For test set $\mathcal{T}_{\in}$, because final move is random, we only test the value error on these examples (rightmost).

- A publicly available $9 \times 9$ Hex game set produced by MoHex 2.0 in relatively strong setting (Gao et al., 2018a). In such configuration, due to extensive H-Search and inferior cell pruning, the player is already close to optimal and it can usually conduct several hundred thousand simulations of MCTS on $9 \times 9$ board size. The test set contains 149362 examples. We call this test set $\mathcal{T}_1$. We use this set of examples to test the policy-head by measuring the move prediction accuracy, state- and action-value heads by measuring the mean squared error (MSE).

- A set of randomly sampled but perfect labeled (using the solver in `benzene`) game states. This set contains 8485 examples. We call this set $\mathcal{T}_2$. This dataset is only used to measure the MSEs of state- and action-value heads, because measuring move prediction accuracy on moves produced by random player is meaningless.

## 4.4 RESULTS

To distinguish, we use AlphaZero-2HNN and AlphaZero-3HNN for AlphaZero learning with two- and three-head architecture respectively. We first use the following parameter choices:

- The maximum number of MCTS simulations per move is set to $n_{mcts} = 800$ for both AlphaZero-2HNN and AlphaZero-3HNN.

- For in-tree selection, as suggested in Gao et al. (2017a; 2018b), we use the default method that uses a weighted combination of prior probability of neural net, RAVE and UCT exploration term, where the weight for prior probability $c_{pb} = 2.47$ (served as $c_{puct}$ when PUCT formula is used).

- Dirichlet noise parameter $\alpha = 0.03$ as AlphaGo Zero and AlphaZero for Go.

- We set move selection dithering threshold $\eta = 10$ due to the short game length in $9 \times 9$ Hex.

- For AlphaZero-3HNN, we use the default expansion threshold $n_{th} = 10$; for AlphaZero-2HNN, $n_{th} = 0$ as in AlphaZero.

We then run AlphaZero-2HNN and AlphaZero-3HNN both for 100 hours. Figure 2 shows the comparative results. For policy learning, both methods achieved similar prediction accuracy on the game examples ($\mathcal{T}_1$) produced by *strong* search algorithms. For the value learning, AlphaZero-3HNN learned much faster than AlphaZero-2HNN both on games states produced by strong player ($\mathcal{T}_1$) as well as examples produced by random play ($\mathcal{T}_2$). On $\mathcal{T}_2$, there is gap between the accuracy of action-value and state-value heads; this suggests that the action-value learning is correlated to policy learning.

From Figure 2 we also observe that one iteration AlphaZero-2HNN is about 5 to 6 times slower than AlphaZero-3HNN. The reason is that the MCTS-2HNN builds a much deeper search tree than MCTS-3HNN (that uses default expand threshold of 10) using the same 800 simulations. We therefore conduct a second run using the following parameter specifications:

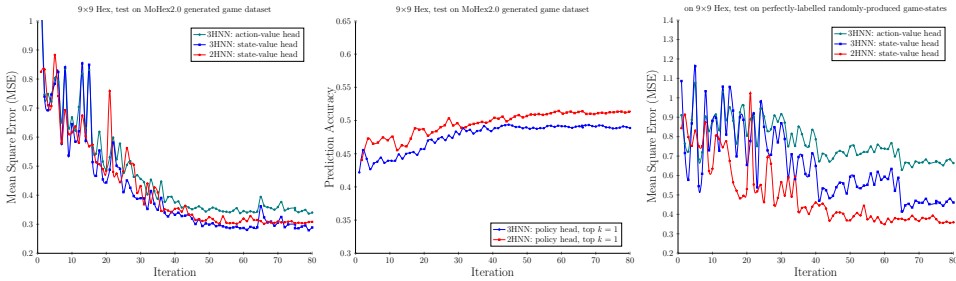

Figure 3: AlphaZero-2HNN versus AlphaZero-3HNN. The only difference is that MCTS-3HNN used $n_{mcts} = 800$ with default expand threshold of 10, while MCTS-2HNN is with $n_{mcts} = 160$ with expand threshold of 0. AlphaZero-3HNN and AlphaZero-2HNN respectively took 90 and 97 hours to complete 80 iterations.

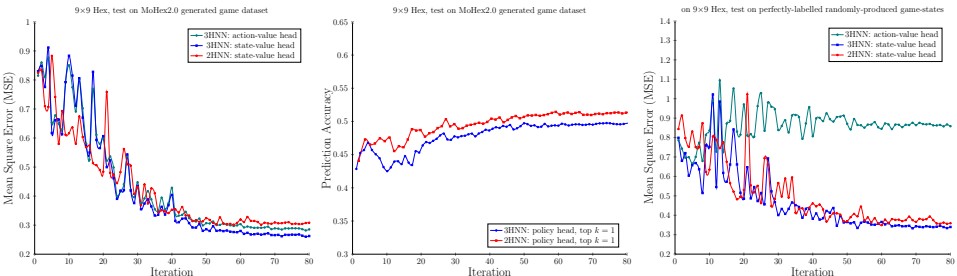

Figure 4: AlphaZero-2HNN versus AlphaZero-3HNN. $R2$ in 3HNN is removed.

- We reduce $n_{mcts}$ to 160 to reduce computation time in each iteration.
- To further encourage exploration, we use move selection dithering threshold $\eta = 30$.
- We change the in-tree selection formula to PUCT and set $c_{puct} = 1.5$ according to (Tian et al., 2019).
- We set the Dirichlet noise parameter $\alpha = 0.15$ as we observe that the branching factor of $9{\times}9$ Hex is about 5 times smaller than $19{\times}19$ Go.

Figure 3 (in red) shows the results obtained by AlphaZero-2HNN learning. In this setting, AlphaZero-2HNN learned much better state-value head: it achieved better results than the AlphaZero-3HNN and AlphaZero-2HNN in Figure 2. However, AlphaZero-2HNN in Figure 3 was using search parameters that are largely different from the AlphaZero-3HNN in Figure 2. To have fair comparison, we then train AlphaZero-3HNN using the same in-tree formula and $c_{puct} = 0.15$, $\eta = 30$ and $\alpha = 0.15$, and include the result to Figure 3. AlphaZero-3HNN did not obtain better predictions than AlphaZero-2HNN on both test sets. For the move prediction accuracy, we found that while the top $k = 1$ accuracy of 3HNN is lower, for $k = 9$, 3HNN and 2HNN have almost identical accuracy of 91%. Since the moves of MoHex2.0 have no guarantee of being optimal, it is still not clear if the 2HNN models are stronger.

We suspect that the reason for the worse value prediction of 3HNN is that the $R2$ term in equation 3 assumes the player is always taking the *best* move in the training games. This does not hold as there is a dithered selection ($\eta = 30$). To see the effect of $R2$, we conduct another training of AlphaZero-3HNN using the same parameters as in Figure 3 except that $R2$ is removed. The obtained result and its comparison to AlphaZero-2HNN is shown in Figure 4, which shows that the state-value head of 3HNN converged to smaller errors as measured on $\mathcal{T}_1$ and $\mathcal{T}_2$. The side effect is that, on $\mathcal{T}_2$, the gap between state- and action-value predictions increased.

### 4.5 MATCH PERFORMANCE

We now directly compare the strength of MCTS-2HNN and MCTS-3HNN using the models in Figures 3 and 4.

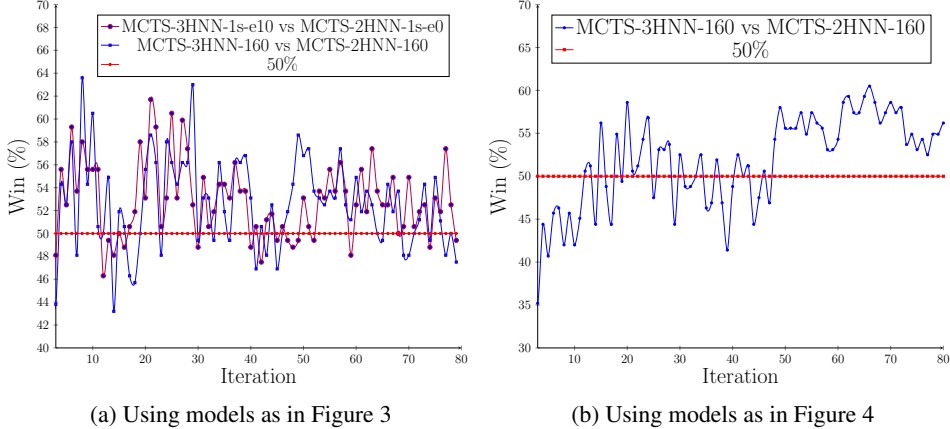

(a) Using models as in Figure 3       (b) Using models as in Figure 4

Figure 5: Match results of 3HNN vs 2HNN. MCTS-3HNN-1s-e10 means MCTS-3HNN using 1s per move with default expand threshold of 10. MCTS-2HNN-1s-e0 means using 2HNN with 1s per move with expand threshold of 0. MCTS-2HNN/3HNN-160 means using 2HNN/3HNN with 160 simulations and expand threshold of 0. Taking the last 20 iteration results, we report the mean win-rates for MCTS-3HNN-1s-e10, MCTS-3HNN-160 (left figure), and MCTS-3HNN-160 (right) are respectively $52.73 \pm 1.04$, $51.25 \pm 1.05$ and $56.54 \pm 0.83$, with $95\%$ confidence. For MCTS, as in training, we use $c_{puct} = 1.5$ but set $\eta = 0$.

Following a common practice (Anthony et al., 2017; Gao et al., 2018b; Anthony et al., 2019; Huang et al., 2013) in Hex, we conduct the match by iterating all opening moves. Thus, each match consists of 162 games (for each opening cell, play 2 games with each player starts first and second with no *swap* rule). Figure 5 contains the results of 3HNN when competing against 2HNN; it shows that, overall, MCTS-3HNN defeated MCTS-2HNN. In Figure 5a, MCTS-3HNN-e10 seems obtained better result than MCTS-3HNN-160; this discovery is consistent with the result reported in (Gao et al., 2018b): delayed node allows more simulations of MCTS and consequently it can lead to better move selection for playing. For Figure 5b, the advantage of 3HNN became visible after around iteration 50, consistent with Figure 4. Note that the test by iterating all openings tend to skew the win-rate to $50\%$ because some openings are very weak or very strong for the first-player: for Figure 5b, if taking only 5 openings ($a2 \ldots a6$) [2] for the test, the mean win-rate of MCTS-3HNN-160 becomes $60.46 \pm 4.37$ ($95\%$ confidence).

Finally, we test our final iteration-80 models of 3HNN and 2HNN by playing against MoHex 2.0. The same as in (Anthony et al., 2019), we let MCTS-2HNN and MCTS-3NN use $n_{mcts} = 800$, and MoHex 2.0 use $n_{mcts} = 10000$. MCTS-3HNN and MCTS-2HNN respectively achieved mean win-rates $91.57 \pm 1.8$ and $89.7 \pm 3.317$ (both with $95\%$ confidence); see Table 2. In comparison, the Zero implementation PGS-EXIT in (Anthony et al., 2019) achieved a win-rate of $58\%$ against MoHex2.0-10000 for 800-simulation search. This indicates the high quality of our AlphaZero implementation for both 3HNN and 2HNN.

## 5 RELATED WORK

The high demand on computation makes reproducing the result or studying the behavior of AlphaGo Zero and AlphaZero significantly difficult. Meanwhile such studies are highly relevant, because the understanding for most parts of these algorithms is still in intuition-level. Following research projects to such end include Leela Zero (Pascutto, 2018), and ELF OpenGo (Tian et al., 2019) — they either use crowd-source for computation or have accessible to thousands of GPUs for parallel selfplay game generation. Some discoveries reported in (Tian et al., 2019) are: (1) the final performance is almost always bounded by the neural network capacity, i.e., improvement can always be achieved by increasing neural network size as more games are accumulated; (2) the PUCT parameter in PV-MCTS seems important; (3) learning (strength of PV-MCTS) has high variance even in

---

[2] Each of these openings took more than a month to solve (Pawlewicz & Hayward, 2013).

the later stage; (4) *selfplay* Elo (Elo, 1978) improvement does not always correspond to real playing improvement of the MCTS player — the monotonic increase of Elo score did not capture the fluctuation in strength; and (5) an important tactic called "Ladder" in Go was never fully master even by increasing MCTS iterations.

AlphaZero style learning has also been studied in games other than Go, chess and Shogi. In particular, in the game of Hex, Anthony et al. (2017) proposed an expert iteration paradigm, applied to $9\times9$ Hex, and obtained a PV-MCTS player that is able to defeat 2011 version of MoHex (Arneson et al., 2010a); the subsequent work (Anthony et al., 2019) explored other search algorithms in expert iteration framework, though on $9\times9$ Hex these algorithms did not produce results better than an AlphaZero implementation that uses MCTS as the search algorithm. Using a policy network for move selection and value network for move evaluation, Takada et al. (2019) proposed a minimax search algorithm and applied it to the game of Hex; starting from random weights, by training the neural network solely using search-based reinforcement learning for a few months, the final minimax-search based player DeepEzo is able to defeat MoHex 2.0 on $13\times13$ in a large margin with 30 seconds per move. However, in formal tournament competition, DeepEzo failed to defeat recent version MoHex3HNN that uses a three-head neural network whose weights were trained on games produced by MoHex 2.0 as well as MoHex3HNN selfplay; see (Gao et al., 2019). Model-free reinforcement learning have also bee applied to the game of Hex, but they did not produce competitive playing strength with search algorithms; see (Young et al., 2016; Gao et al., 2018a). In the game of NoGo, Lan et al. (2019) explored the idea of dynamically increasing neural network size during the process of AlphaZero training, and showed that a Monte Carlo tree search with multiple small and large networks could accelerate the learning.

## 6 CONCLUSIONS

Using the game of Hex as an object, we have presented an empirical study on synchronous AlphaZero learning with two- and three-head neural networks. Our results suggest that the three-head network is indeed compatible with the zero-style closed loop learning paradigm, consistent to the conjecture made in (Gao et al., 2018b); moreover, in the game of Hex, the new architecture also enabled faster search and yielded better performance.

On the other side, the adjustment we have made on search for using the new action-value head is rather limited: only a non-zero expansion threshold is adopted to take advantage of the available action-value estimates. Other opportunities, such as combining the estimates of state-value and action-value heads, could be studied. Furthermore, it is not clear if MCTS is the best choice for search at all. Unlike $\alpha\beta$, MCTS does not directly apply hard-min and -max upon estimated value functions which tends to amplify the approximation error. Early research (Nau, 1982; Pearl, 1983) have also suggested some alternative methods that do not have such issues. For example, the product propagation (PP) (Pearl, 1983) have long been conjectured as a promising alternative (or better than?) to minimax, while in practice PP requires to estimate the probabilities of win/lose which are not instantly available. Previous studies (Kaindl et al., 2015) typically normalize a static evaluation function into probabilistic estimates, and in practice if PP is superior to minimax is unclear. It remains to experimentally verify if neural networks could be used for provide better probabilistic estimates.

Playing and solving algorithms for two-player games have seen separate developments. While early work in playing majorly focus on more memory-efficient depth-first procedures that use $\alpha$ and $\beta$ bounds for pruning, the development of solving algorithm (e.g., proof number search (Allis, 1994)) quickly switched to best-first paradigm as they are more tolerant to heuristic estimates backpropagated from leaf evaluations. MCTS is a best-first search algorithm that has been extensively used for playing but not solving, while proof number search (Allis et al., 1994) variants (e.g., (Gao et al., 2017b)) are good at fast-solving but do not perform well at playing games. Unifying playing and solving algorithms that can fast solve game states given sufficient computation time and memory while also can provide high-quality moves given limited computation budget is worth further investigating. More preferably, such an unified algorithm should be compatible with AlphaZero style learning, therefore enabling improved performance in both playing and solving as neural network learning progresses on.

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

# A MONTE CARLO TREE SEARCH WITH DELAYED NODE EXPANSION

Monte Carlo tree search (MCTS) is a best-search algorithm that repeatedly conducts the following steps:

1. Selection. Given an existing search tree and a single starting node $s_R$, a leaf node $s_L$ in the tree is selected by a series of action-selection starting from $s_R$. an action,

2. Evaluation. The leaf node $s_L$ is evaluated, either by Monte Carlo playout or an external evaluation function (e.g., a neural network).

3. Expansion. The leaf node $s_L$ may be expanded and if so, all child nodes of $s_L$ will be *created* and added to the search tree.

4. Backpropagation. The evaluation of $s_L$ obtained at step 2. is backpropgated to the search tree.

A complete execution of these steps is called one simulation (or iteration) of MCTS. Delayed node expansion refers to an implementation of step 3. where the leaf node $s_L$ will be expanded only when it has been visited at least $n_{th} > 0$ number of times. Conversely, the non-delayed expansion implementation expands $s_L$ when it has been selected the first time, i.e., equivalently, $n_{th} = 0$.

A three-head network can be used for delayed node expansion, because every leaf node in the search tree contains an value estimate that was stored upon its *creation*.

# B IMPLEMENTATION DETAILS

For understanding convenience, we document how to configure the MCTS in MoHex for various use. Most of them have been implemented in `benzene` except that those involving neural networks and AlphaZero are newly added.

## B.1 MOHEX TURN OFF KNOWLEDGE COMPUTATION

```
param_mohex perform_pre_search 0
param_mohex knowledge_threshold "999999"
```

## B.2 SETTING IN-TREE SELECTION FORMULA

Two formulas have been implemented, the RAVE formula:

$$(1 - \omega)\left(Q(s,a) + c_b\sqrt{\frac{\ln N(s)}{N(s,a)}}\right) + \omega R(s,a) + c_{pb}\frac{p(s,a)}{\sqrt{N(s,a)+1}}$$

, and a PUCT formula:

$$Q(s,a) + c_{pb}P(s,a)\frac{\sqrt{\sum_b N_r(s,b)}}{1 + N_r(s,a)}.$$

They can be configured using the following commands.

```
param_mohex use_rave 0
param_mohex progressive_bias 1.5
param_mohex uct_bias_constant 0.0
param_mohex root_dirichlet_prior 0.15
param_mohex moveselect count
param_mohex moveselect_ditherthreshold 30
param_mohex extend_unstable_search 0
```

If `use_rave` is true, a in-tree selection using RAVE will be used, otherwise the PUCT formula will be used. $c_b$ is set by `uct_bias_constant`, and $\omega$ is a dynamic RAVE weight. The prior probability $P(s,a)$ will be set to $P(s,a) \leftarrow (1 - \epsilon)P(s,a) + \epsilon\text{Dir}(\alpha)$ where $\epsilon = 0.25$ if `root_dirichlet_prior` $\neq 0$, otherwise no dirichlet noise will be used.

### B.3 MoHex TURN OFF PLAYOUT

```
param_mohex use_playout_constant 0.0
```

By such, MoHex solely uses neural net estimates for leaf node evaluation.

### B.4 MoHex LOAD NN MODELS

```
nn_load path_to_nn_model
```

By such, MoHex can load a trained neural net model into its search.

### B.5 MoHex SETS NUMBER OF ITERATIONS

```
param_mohex max_games 800
```

By such, MoHex will conduct an 800-iteration search.

### B.6 MoHex SETS EXPAND THRESHOLD

```
param_mohex expand_threshold 0
```

By such, MoHex will expand a leaf at its first visit.

### B.7 NEURAL NET TRAINING

The neural network was trained using stochastic gradient descent with momentum $0.9$. The batch size is $128$, initial learning rate is $0.005$, with an annealing factor $0.9$ each iteration until a minimum learning rate $0.000005$ is reached. Note that our initial learning rate is order of smaller than that of AlphaZero because our batch-size is 32 times smaller. At each iteration, we train the neural network for 5 epochs and then pass the obtained model to MCTS workers for next iteration game generation. $L_2$ regularization is set to $c = 0.0001$.

The neural network we used is exactly the same as in (Gao et al., 2018b), containing 10 residual blocks, each with 32 filters.

### B.8 MCTS SELFPLAY

We use 60 parallel MCTS workers for selfplay, each producing 200 games each iteration. We reuse the subtree in each selfplay game (by configuring `param_mohex reuse_subtree 1`). Unlike Go, Shogi and Chess, for $N \times N$ Hex, the maximum game length is $N^2$, therefore we do not specify a resign threshold during selfplay, either maximum game length. To accommodate the *swap rule* in Hex, for each selfplay game, Black starts first by uniformly sampling a cell on the board.

### B.9 COMPUTATION HARDWARE

We use the same hardware for neural network training and selfplay game generation. Specifically, the computer has an Intel(R) Xeon(R) CPU E5-2690 v4 @ 2.60GHz with 56 processors with 500 GB RAM, and 6 Tesla P100 GPUs each with 16 GB RAM. During selfplay, the parallel workers are evenly distributed to the available GPUs.

### B.10 TURN OFF DATA AUGMENTATION IN 3HNN WHEN SELECTED MOVE IS NOT THE BEST

The data augmentation error in Eq. 3, i.e., $\frac{\max(-z_s,0)}{|\mathcal{A}(s)|} \sum_{a' \in \mathcal{A}(s)} (z_s + q(s, a'))^2$, assumes that in the selfplay game both players are selecting the *best* actions produced by search. This assumption does not hold when dithering threshold $\eta > 0$. Instead of just removing $R2$, anther way is to remove $R2$ in training 3HNN when the move played in the example game is not with the highest move probability. This scheme was not tested in the paper.

Table 2: Detailed match results against MoHex 2.0. Each set of games were played by iterating all opening moves; each opening is tried twice with the competitor starts first and second, therefore each match consists of 162 games. We use the final iteration-80 models for 2HNN and 3HNN from Figure 4. The overall results are calculated with 95% confidence. MCTS-2HNN and MCTS-3HNN used 800 simulations per move with expand threshold of 0. MoHex 2.0 used default setting with 10,000 simulations per move.

| Player \ Set | 1 | 2 | 3 | Overall |
|---|---|---|---|---|
| MCTS-2HNN-800 | 86.4% | 92.0% | 90.7% | $89.7\% \pm 3.32$ |
| MCTS-3HNN-800 | 93.2% | 91.4% | 90.1% | $91.57 \pm 1.77$ |

### B.11 MATCH CONFIGURATION

For MoHex 2.0, we use the default setting but restrict to 10000 iteration MCTS, which is equivalent to the following:

```
param_mohex use_rave 1
param_mohex uct_bias_constant 0.22
param_mohex progressive_bias 2.47
param_mohex max_games 10000
```

For MoHex with 2HNN or 3HNN, we use the following:

```
param_mohex use_rave 0
param_mohex uct_bias_constant 0.0
param_mohex progressive_bias 1.5
param_mohex max_games n_mcts
param_mohex use_playout_constant 0.0
```

The difference for 2HNN and 3HNN is that, for 2HNN, we always set `param_mohex expand_threshold 0`, while for 3HNN, other non-zero thresholds are also used as specified in the context. For the test between 2HNN and 3HNN, `n_mcts=160`. For test against MoHex 2.0, `n_mcts=800`.

## C VISUALIZATION

We show an visualization of the learned policy and action-value heads using HexGUI [3].

---

[3] accessed by `https://github.com/ryanbhayward/hexgui`

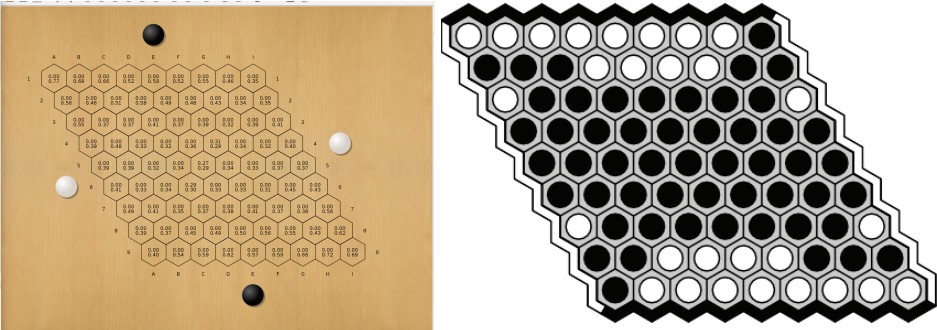

Figure 6: The left subfigure is obtained by inferring the 3HNN model from iteration 80 as in Figure 4. The right subfigure is the ground truth of each cell (black means first player win, assuming black plays first). For each cell, the upper number is the prior probability from policy head, the lower number is the action estimate from action-value head. Note that the action-value is with respect to the player to play after taking that action. The policy head provided non-zero probabilities at only three strongest central openings, while the action-value head provides a set of estimates that are mostly consistent with the ground truth of each opening move.

