# OpenReview forum: "Three-Head Neural Network Architecture for AlphaZero Learning"
_ICLR.cc/2020/Conference — Reject_

### Official Review · AnonReviewer3 · 2019-10-12
**Official Blind Review #3**

**Rating:** 6

**Review:**

The paper applies three-head neural network (3HNN) architecture in AlphaZero learning paradigm. This architecture was proposed in [1] and the paper builds upon their work. In AlphaGo and AlphaZero 2HNN is used, which predicts policy and value for a given state. 3HNN also predicts action-value Q function. In [1], the three new terms are added to the loss to train such a network, and the network is trained on a fixed dataset. The paper utilizes the same 3HNN idea with the same loss, and the contribution is that 3HNN is trained synchronously with MCTS iterations on an updating dataset (“AplhaZero training style”). Learning speed of 3HNN is shown to be higher than that of 2HNN. The special attention is drawn to varying the threshold expansion parameter, as the 3HNN architecture allows to set it above zero, while 2HNN does not. The approach is demonstrated on the game of Hex. Results are presented on two test datasets: positions drawn from a strong agent’s games and random positions. Labels in both datasets are perfect, obtained by a special solver.

I tend to reject the paper, because the demonstrated results suggest that the models were not tuned well enough. Indeed, the paper claims that the parameters were not tuned. The paper claims using threshold expansion > 0 to be one of the main advantages of 3HNN. However, the best model in the experiment is the one with the parameter equals zero. Overall, for a purely experimental paper, the experiments are too crude.

Main argument
1.	It seems that some of NN models didn’t learn at all:
           • 	Figure 2, 2HNN model. MSE on both the left and right plots is not improving. Moreover, on the right plot it
                fluctuates around 1, which is the performance of a random guess.
           •	Figure 3, right. MSE of the 3HNN model is not improving. Probably, random positions are too unnatural and
                nothing similar is presented in the dataset drawn from MCTS iterations.
2.	Supmat reveals, that some of the models are in fact learned using the different loss than it is said in the paper. In particular, data augmentation term is sometimes on and sometimes off. A disabling scheme is suggested, depending on the dithering threshold and the number of the moves played before the state s. Some models use the scheme, for others the term is always on. This should be clearly stated in the main text, not in the supmat.
Also, there is an experiment in the supmat, when the data augmentation term is always off. The influence of this term is itself interesting, as it is one of the reasons 3HNN is learning q-function at all. However, introduction of this term itself is the contribution in [1]. I suggest to add to the main part of the paper the experiment, comparing three regimes: 1) with the scheme, 2) term always on and 3) term always off. In fact, it is almost done, as all three regimes are used in different figures, but somewhy the final comparison (with other parameters fixed) is not shown and partly concealed in the supmat. It could become methodological improvement of the paper over [1].
3.	When the leaf node s is expanded, the v function of the new node s’ = s ∪ a is initialized to predicted q(s, a). It is one of the advantages of 3HNN and allows node expansion threshold. When s’ itself is expanded, how do you merge q(s,a) with backup values during mcts iterations?
4.	Nothing is said about the architecture of NNs. If the representation of the state is the same as in [1], it should at least be mentioned.
5.	How exactly does figure 2 shows that one iteration AlphaZero-2HNN is 5-6 times slower, than AlphaZero-3HNN. Probably the definition of data point in figure 2 is missing (e.g. one data point corresponds to one MCTS iteration).
6.	Figures 3 and 4 basically shows the same experiment, but with different models: AlphaZero-2HNN versus AlphaZero-3HNN with threshold 0 on Figure 3 and AlphaZero-2HNN versus AlphaZero-3HNN with threshold 1 on Figure 4. They could be united in one figure.
7.	Results from Figures 3 and 4 suggest, that 3HNN with threshold = 0 is better, than with threshold = 1. However, the paper claims setting threshold > 0 as an advantage. If it allows to save time (as each mcts iteration is faster), maybe they should be compared plotting time on x-axis (like in Figure 2)?
8.	Please provide error bounds in table 2.

Additional arguments
Argumentation presented in this section didn’t affect the score. However, it might improve the paper.
1.	3HNN predicts q(s,a), which should be equal to v(s’), where s’ = s ∪ a (state after action a is taken in state s). It would be interesting to see how condition q(s,a) == v(s ∪ a) holds during  3HNN models learning. It can be checked on a first dataset (drawn from games), or a special random dataset, containing consecutive positions, could be generated. Probably, this condition could potentially be an additional loss term.
2.	Also, it would be interesting to see how the condition between p and q holds. The higher q(s, a), the lower should be p(a). There is an interesting illustrating figure 7 in supmat, it may be presented in the main text. Also, it could be interesting not only for the first move.
3.	The supmat claims that the motivation for turning off data augmentation term is that it assumes that in the selfplay game both players are selecting the best actions produced by search. Is it connected with the fact, that in the game of Hex all states are either winning or losing, according to theorem proved by Nash? How does this term would work for games with a draw trend, for example chess? In chess, for a lot of states the “ground truth” v(s) would be close to zero and there is no action guaranteeing the win.
4.	The paper claims “Elo (Elo, 1978) improvement does not always correspond to real playing improvement—the monotonic increase of Elo score did not capture the fluctuation in strength”. Citation needed, what fluctuation in strength is not captured? Is it specific to game of Hex? For example, Elo is used as the main measure of total agent strength in AplhaZero papers, as well as by chess community (both chess programs and human players).

Minor comments
1.	Page 4, section 2.2: Even though ... . Our summarization -> Even though ... , our summarization.
2.	Page 4, table 1: mvoe -> move.
3.	Page 7, bullet point above section 4.4: perfect -> perfectly.
4.	Page 7, the lowest paragraph. “For the value learning, however, due to fast search, the AlphaZero-3HNN learned much faster than AlphaZero-2HNN both on games states produced by strong player (T1) as well as examples produced by random play (T2).”
It is confusing, it seems that 3HNN on datasets T1 and T2, however, it was only tested on these datasets.
5.	Page 8: imposing an an auxiliary task -> imposing an auxiliary task.
6.	Page 9: “produce playing strength significantly stronger than” – reformulate.

[1] Chao Gao, Martin Müller, and Ryan Hayward. Three-head neural network architecture for monte carlo tree search. In IJCAI, pp. 3762–3768, 2018b.

=====Post Rebuttal=====
Score updated from 3 to 6.


**Experience Assessment:**

I do not know much about this area.

**Review Assessment: Checking Correctness Of Derivations And Theory:**

N/A

**Review Assessment: Checking Correctness Of Experiments:**

I carefully checked the experiments.

**Review Assessment: Thoroughness In Paper Reading:**

I read the paper thoroughly.

---

> ### Author Response · Authors · 2019-11-15
> **Thanks for the useful comments, we have addressed the referee's concerns in the revision**
>
> Thank you for the referee's useful comments.
>
> We think the revised version has fixed the referee's concerns. Missing evaluations in the initial submission have now been added, and now we only focus on two cases for 3HNN, 1) expand threshold 0, 2) default expand threshold of 10.
>
>
> Answers to the major concerns.
> 1. Parameters well-tuned for 2HNN and 3HNN?
>       It is difficult to say which parameter choice is the "well-tuned" and which is not.
> We agree that from Figure~2, it seems for both 2HNN and 3HNN, the hyperparameter choices were not well-set. We include this Figure mainly because this parameter setting was a natural choice by the supervised learning results in (Gao et al. 2017, Gao et al. 2018b).
>
> However, we believe this is not be the case for Figures 4 and 5, as clearly shown in the picture.
>
> Another fact is that, in our evaluation, our 2HNN and 3HNN models in Figures 5 finally achieved 90% win-rate (with 95% confidence); see Table 2.  By comparing to another AlphaZero implementation for 9x9 Hex (see (Thomas et al. 2019)), we believe that our AlphaZero for both 3HNN and 2HNN have yielded  high quality neural nets, which is not possible if the hyperparameter choice is poor.
>
> 2. Whether with R2 or without R2?
> To be clear, in the revised version, we keep only two variants for 3HNN.
>  (a) always with R2; and (b) always no R2.
>
> We added a direct head-to-head playing, which shows that in both cases, 3HNN in MCTS yielded stronger playing than 2HNN.
>
> 3. How to merge q(s,a) and v(s) when s is expanded?
>  A few possibilities have been suggested in (Gao et al. 2018b). Here, to keep it simple,
> we only back up v(s) if expanding s. This is also the default choice in (Gao et al. 2018b).
>
> 4.  Architecture of NN?
> Yes, we are using exactly the same architecture as in (Gao et al. 2018b). This has been clarified in the revision.
>
> 5. Why one iteration of 2HNN is slower than 3HNN in Figure 2?
> This is because for the same 800 simulation search, MCTS-3HNN used default expansion threshold of 10, while MCTS-2HNN used 0.  This makes them producing a game with different speed.  MCTS-3HNN took about 1s per move, while MCTS-2HNN took up to 10s per move.
>
> 6 & 7. Figures 3 and 4?
> We have revised these figures.
>
> 8. Error bound in Table 2?
>  We have revised Table 2 and included error bounds. It shows that with 95% confidence, our players achieved ~90% win-rates.
>
> --------------------------------------------------------------------
> Answers to "suggestions to improve the paper":
> 1 & 2.  There are many can be investigated. We aim to keep it simple in this paper.
>
> 3. No, it not related to the theorem of Nash. The technique itself does not prevent its usage in games where are draws. In Figure 4, we see 3HNN worked well even without it.
>
> 4. The citation was there. This was from (Tian et al. 2019).
> To be clear, we have rephrased it to "selfplay Elo".  The argument is, for example,  if A player is beaten by B, B beaten by C. If A has Elo 100, B may get 150, then C 200, but in real-playing C may not be really stronger than A, or very likely, not 100 Elo stronger than A.
>
> --------------------------------------------------------------------
>  Answer to minor comments:
>  Thanks for the careful reading, we have fixed these writing errors.

---

### Official Review · AnonReviewer2 · 2019-10-23
**Official Blind Review #2**

**Rating:** 3

**Review:**

The paper proposed to use three-head network for AlphaZero-like training. The three-head network is used to predict policy, value and q function after an action is taken. While three-head network is presented by a prior work [1] and is learned via supervised learning on a fixed dataset, this paper mainly applies it to AlphaZero training for the game of Hex 9x9 and shows preliminary results.

While the idea is interesting, there are many issues in the experiments and the conclusion is quite indecisive. So I feel that the paper is not ready for publication yet and thus vote for rejection.

Why we need expansion threshold n_th to be 10? If you keep visiting the same node without expansion, won’t the same node back-propagate the same value (or q) 10 times before expansion? If that’s the case, what’s the difference if we just back-propagate once? Note that if n_th = 0 then prediction of q(s, a) is no-longer necessary (except that predicts q(s, a) becomes an aux task during training, as mentioned in the caption of Fig. 3).

Fig. 2 shows that 3HNN trains faster than 2HNN. However, it looks like 2HNN and 3HNN show drastically different training curves, and are probably operating at different regions. In the text, the authors also acknowledge that one iteration of 2HNN is 5-6 times slower than 3HNN, since 2HNN builds a much deeper search tree. This bring about a question: is the performance difference due to unfavorable hyper-parameters on 2HNN (or other factors)? The paper doesn’t answer that.

The text claims that when n_th = 0, 3HNN performs better than 2HNN, however, the figure shows that 2HNN has lower or comparable MSE than 3HNN. The prediction accuracy is better, though. When n_th = 1, Fig. 4 shows that the 2HNN is doing comparable or better in terms of MSE and Prediction Accuracy than 3HNN (compared to perfect play). This somehow defeats the purpose of using the third head of q(s, a) that only helps when n_th > 0.

In Table 2, do you have standard derivation? Note that AlphaZero training is not that stable and the performance (in particular the initial performance since the performance might take off earlier or later) against a known bot can vary a lot, the difference between 56% and 63% can be purely due to noise. Also, how is the resulting model compared against MoHex-3HNN [1] and MoHex-CNN [2]? Note that MoHex-3HNN [1] shows 82.4% over MoHex 2.0 on 13x13, but is trained supervisedly, and Table 2 shows slightly better performance. So I am curious their performance comparison.

Minor:
The term “iteration” seems to be defined twice with different meanings. It is defined as one MCTS rollout (see Appendix A) and also defined (in Fig 1) as one full synchronization of self-play and training (AlphaGo Zero setting). This causes a lot of confusions. I believe each dot in Fig. 2 is “iteration” in the AlphaGo Zero sense.

Finally, although many hardware information is revealed in the appendix, maybe it would be better if the authors could reveal more details about their AlphaZero-style training, e.g., how long does it take for each move and for each self-play game? How long does it take to wait until all self-play agent returns all games? Is there any synchronization overhead? This could give the audience some idea about the computational bottleneck.

From the current number, it seems that 60 self-play processes are run on 56 cores, and each AlphaGo iteration takes approximate 5 hours (read from Fig. 2) with 200 games per self-play process. Assuming there is no synchronization overhead and 1 core per self-play process, this yields 200 games/5 hours per core, which is 1.5 min (or 90s) per game. Since each game has 9x9 = 81 moves, this means that it costs ~1.1 sec per move. Is that correct?

[1] Chao Gao, Martin Muller, and Ryan Hayward. Three-head neural network architecture for Monte Carlo tree search. In IJCAI, pp. 3762–3768, 2018.

[2] Chao Gao, Ryan B Hayward, and Martin Muller. Move prediction using deep convolutional neural networks in Hex. IEEE Transactions on Games, 2017.

=====Post Rebuttal=====
I really appreciated that the authors have made substantial efforts in improving the paper and adding more experiments. However, the additional change makes the story a bit more convoluted. After substantial parameter tuning on the 2HNN side, It seems that 3HNN is only slightly better than 2HNN (Fig. 5 in the revision, > 50% winrate, but it is not clear how much ELO it is better). Unfortunately, after tuning, 2HNN actually shows comparable performance in terms of speed (updated Fig. 3 and 4, middle columns), which somehow tarnishes the claims of the paper that 3HNN is better than 2HNN.

The final performance against 10000-rollouts MoHex2.0 is 89.7% (2HNN) versus 91.6% (3HNN), so the performance is slightly better with 3HNN. This number is much better than previous works e.g., PGS-EXIT (Thomas et al. 2019). This indeed shows that the paper does a good job in terms of engineering and performance push (agreed with R1). In my opinion, the paper can be better rewritten as a paper that shows strong performance in Hex, compared to previous works, plus many ablation analysis.

I keep the score.

**Experience Assessment:**

I have published in this field for several years.

**Review Assessment: Checking Correctness Of Derivations And Theory:**

I carefully checked the derivations and theory.

**Review Assessment: Checking Correctness Of Experiments:**

I carefully checked the experiments.

**Review Assessment: Thoroughness In Paper Reading:**

I read the paper thoroughly.

---

> ### Author Response · Authors · 2019-11-15
> **we have updated the paper, sincerely wishing the referee to reconsider the value of the updated version**
>
> We thank the referee for the constructive comments.  Yes, this work is mainly an empirically study of 3HNN in AlphaZero style training.
> This study is difficult because of (1) the computation demand, (2) the large number of hyperparameter choices.   Due to time constraint, some evaluations were not added in our initial submission. We have now added these results to showcase 3HNN.
> We hope this revision could make the referee to reconsider the value of this work.  We wish that this work could inspire a broader discussion of 3HNN and eventually lead to a larger scale study.
>
> Answers to the major concerns
> 1. Why non-zero expansion threshold?
> This has been explored in (Gao et al 2018b). The role of delayed node expansion is to accumulate more simulations given the same computation resource.  In our MCTS selfplay,
> with default expansion threshold of 10, it generally took over 1000 simulations per move.
> With expansion threshold of 0, it took ~200 simulations per move.  (one simulation is one iteration of MCTS; in some places, it is also called rollout/playout).
>
> In (Silver et al. 2017b), it is argued the merit of MCTS, in comparison to Alpha-Beta, is that it uses average to achieve stable evaluation. Clearly, delayed node expansion leads to more averaging than instant expansion.
>
> In the revised paper, we only investigate two possibilities, either 0 or the default expand threshold of 10. See also our response to Rev1.
>
> 2.  Unfavourable parameter choice for 2HNN?
>   After reducing n_{mcts}=160, and revise \eta, dirichlet noise, 2HNN obtained excellent result; see Figures 3 and 4.
> A cross reference is to the Zero implementation in (Thomas et al. 2019; cited). Both against 10000-simulation-MoHex2.0,
> our implemented MCTS-2HNN achieved over 90% win with a 95% confidence; see Table 2;
> while PGS-EXIT (Thomas et al. 2019) achieved 58%;
> both using 800-simulation for search.
>
> 3. Is 3HNN's performance indeed better than 2HNN?
>   As the MSEs and move prediction accuracy are only surrogate measurements on the strength of the neural network. To see if really 3HNN is stronger than 2HNN, we now present direct head-to-head match result.
> See Figure~5, which shows MCTS-3HNN mostly achieved over 50% win-rate. Each match consists of 162 games. Each curve in Figure~5 was produced by 162*80 games. They were not included in the initial version because they were not finished (taking a week).
>
> 4. Significance of evaluation results with MoHex 2.0?
>
>    We now performed three sets of match, each with 162 games. With 95% confidence, our final MCTS-3HNN achieved $91.57\% \pm 1.8$  while MCTS-2HNN achieved $89.7\% \pm 3.32$, against MoHex2.0.  See Table~2 in the revised paper.
>
> 5. Is the result comparable to MoHex3HNN (Gao et al. 2018b) and MoHex-CNN (Gao et al. 2017)?
>     These programs were only run on 13x13 Hex, while in this paper, we consider only 9x9 Hex.
>
>
> Minor issues:
> 1) To distinguish, we have now replaced "iteration" with "simulation" in describing MCTS.
> 2) Yes. you’re correct. In our log, it shows that for Figures 3 and 4, MCTS-2HNN and MCTS-3HNN took around 1s per move. The neural network training time is negligible in comparison to game-generation. We run 60 workers on a 56-cpu computer, the relative speed of each self-play work is rather similar. We did not observe serious synchronization overhead.

---

### Official Review · AnonReviewer1 · 2019-10-27
**Official Blind Review #1**

**Rating:** 6

**Review:**

This paper applies the three-head neural network architecture as well as the corresponding training loss proposed in (Gao et al., 2018b) to alphazero style learning of the Hex game. The paper is mainly an empirical study, and shows that the architecture leads to faster and better learning results for Hex. The evaluation is done on two datasets, one with examples from near-optimal players produced by MoHex 2.0, and the other from randomly sampled but perfectly labelled examples generated by benzene. Performance improvement is evaluated from several different perspectives, including state-value errors, action-value errors and policy prediction accuracies. Finally, the match performance is also reported for competing with MoHex 2.0, one of the state-of-the-art agent for Hex.

Generally speaking, the paper does a good job in introducing and analyzing the structure of the alphazero learning scheme and the related alphago and alphago zero schemes, and the experiments within the scope of Hex is relatively thorough and the performance improvement is consistent and convincing.

However, the description of the three-head neural network in Section 3 is too brief, and without looking at the original paper (Gao et al., 2018b), it is quite hard to understand the motivation of the objectives (especially the definitions and explanations of R1, R2 and R3).

Additionally, the challenge of applying three-head neural network architecture in the alphazero learning setting is almost not mentioned. In particular, what are the modifications needed compared to the original work (Gao et al., 2018b)? The authors may want to explain clearly how the training scheme is different, and clearly state what the detailed neural network architecture (at least in the appendix) used is, and how they are different from the original alphazero paper and (Gao et al., 2018b). Without these explanations, the significance of the paper would be largely limited to coding and engineering efforts (which are also valuable but not that much in the research sense).

Another related issue of this paper is that it is not clear (at least to me, who know little about the Hex game) how difficult it is to tackle Hex (compared to Go, Shogi and chess, etc.). The authors may want to elaborate more on this as well to further showcase the significance of the work.

Finally, there are also some inconsistency in the hyper-parameter choices and architecture design. In particular, it is not clear why the authors choose the expansion threshold to 0 in the match performance part, whereas the authors use threshold 10 elsewhere. The turning on and off of the data augmentation in 3HNN in different experiments mentioned in the appendix are also not well explained.

Nevertheless, I still value the paper's effort and success in applying a newly proposed approach for a relatively challenging real-world game problem, despite the issues about experimental design and writing mentioned above.

Some minor suggestions: the title of the rightmost plots should better be "perfectly labelled examples" instead of "perfect examples", and the authors may want to make it clearer which plot corresponds to dataset T1 and which corresponds to T2.

**Experience Assessment:**

I have read many papers in this area.

**Review Assessment: Checking Correctness Of Derivations And Theory:**

I assessed the sensibility of the derivations and theory.

**Review Assessment: Checking Correctness Of Experiments:**

I assessed the sensibility of the experiments.

**Review Assessment: Thoroughness In Paper Reading:**

I read the paper at least twice and used my best judgement in assessing the paper.

---

> ### Author Response · Authors · 2019-11-15
> **Thank you for the valuable comments, we have updated with a revised version**
>
> Thank you for the valuable comments. We have revised the paper for better exposition.  We summarize and answer the referee's major questions below.
>
> 1. What are the challenges in applying AlphaZero with 3HNN?
>       1) The first challenge is computation resource, which made Gao et al.2018b only evaluate 3HNN on fixed data with supervised learning.
>       2) The second challenge is that it is not clear if the R2 term the loss would work on AlphaZero style training, because this data-augmentation assumes the move selected by search is the "best" among all candidates. AlphaZero sometimes samples a move not with the highest visit count for exploration.
>
> To investigate “the value of the R2 term in the loss”, we present results of two versions, (v1) always with R2; (v2) always without R2.
>
> We have now included direct game between 2HNN and 3HNN in Figure 5 (in the revised paper uploaded to openreview), which shows that when used in MCTS,  3HNN models from (v1) and (v2) both obtained stronger playing than 2HNN.
>
> The prediction of value functions are mixed though. (i) Without R2 it led to better state-value-learning, though its action-head did not generalize well on random game-states, (presumably because of the lack-of-data). (ii) With R2 the action-value head generalized much better than without R2 on random game-states, though the state-value head worse than (v2).
>
>
> [We note that there is a third scheme: remove R2 only when it sees in training that the selected move is not with highest probability.  This one was not investigated.  Previously, we investigated a scheme by removing all R2 before $\eta=30$. But this scheme looks naive. Considering the page limit, the results were removed from the paper. ]
>
> 2. What is the relative difficulty of Hex?
>      In AI, the difficulty of Hex is often compared to Go. In late 1990s, in parallel to Go, researchers began to realize that the advancements developed for chess/checkers (and so on) do not work well on Hex. Van Rijswijck 2002a made a metaphor that Hex should be regarded as a "bee"  if calling chess a "fruitfly" for AI.  For state-space sizes, 9x9 Hex is 10^{38} while chess is 10^{47}, Shogi is 10^{71}. Commonly, 11x11 and 13x13 Hex are used by humans for playing, whose space sizes are 10^{56} and 10^{79}.  Hex is sometimes played at 19x19 board, whose space size is close to 19x19 Go.
>
>     The major difference between Hex and Go is that, many graph-theoretical properties can be identified in Hex; these could be used as an exact knowledge for pruning the state-space size without removing optimal strategy. In Go, such exact knowledge is more difficult to formalize, in part because of the KO rule (there is why there are Japanese, Chinese and other rules). Hex has simpler rules than Go, but as Go, well-playing strategy is hard to describe.
>     For scientifically and empirically studying the behaviour of AlphaZero, we believe that Hex is no less significant than Go. We chose 9x9 Hex because of (1) it is the largest board size where existing solver can be used to label a relatively large set of random game-states (2) it has been used in previous Zero studies (eg. Thomas et al.2017, 2019).
>
> 3. Why expansion threshold 0 or 10?
> The non-zero expansion threshold has been a popular scheme in MCTS for playing games before deep neural nets came to stage (e.g. all MoHex versions):
> -- 10 is the default one used in MoHex.
> -- AlphaGo-Fan also used non-zero expansion threshold, although, with 2HNN, later developed AlphaGo-Zero/AlphaZero changed to 0.
>
>
> 4. Appendix result not well explained?
> To have coherent flow, we have moved appendix result on 9x9 Hex to main text, and removed 8x8 result.

---

### Decision · Program_Chairs · 2019-12-19

**Decision:**

Reject

**Comment:**

The authors provide an empirical study of the recent 3-head architecture applied to AlphaZero style learning. They thoroughly evaluate this approach using the game Hex as a test domain.

Initially, reviewers were concerned about how well the hyper parameters for tuned for different methods. The authors did a commendable job addressing the reviewers concerns in their revision. However, the reviewers agreed that with the additional results showing the gap between the 2 headed architecture and the three-headed architecture narrowed, the focus of the paper has changed substantially from the initial version. They suggest that a substantial rewrite of the paper would make the most sense before publication.

As a result, at this time, I'm going to recommend rejection, but I encourage the authors to incorporate the reviewers feedback. I believe this paper has the potential to be a strong submission in the future.